# Effect of Continuous Administration of Enalapril Maleate on the Oocyte Quality and In Vitro Production of Parthenote Embryos in Nulliparous and Multiparous Goats Undergoing Serial Laparoscopic Ovum Pick-Up

**DOI:** 10.3390/ani9110868

**Published:** 2019-10-26

**Authors:** Pamela A. Bravo, Maria E. Moreno, César C.L. Fernandes, Rafael Rossetto, Camila M. Cavalcanti, Denilsa P. Fernandes, Davide Rondina

**Affiliations:** Faculty of Veterinary Medicine, Ceará State University (UECE), Fortaleza 60714-903, Ceará, Brazil; mvet.pamelabravo@gmail.com (P.A.B.); mveugemoreno@hotmail.com (M.E.M.); rafael.rossetto@hotmail.com (R.R.); camilacavalcanti89@hotmail.com (C.M.C.); denilsas2@hotmail.com (D.P.F.)

**Keywords:** IVP, LOPU, Renin angiotensin

## Abstract

**Simple Summary:**

One of the main obstacles to the in vitro production of embryos in goats is the low ovarian response to hormonal treatments and low oocyte quality. Thus, several strategies have been performed to improve the reproductive performance of goats, including the development of new hormonal protocols as well as the use of other drugs that act directly or indirectly on reproductive function. In this experiment, we tested the use of a hormonal protocol aimed at maximizing the ovarian response and, in parallel, an angiotensin-converting enzyme (ACE) inhibitor was administered daily as an adjuvant. In recent years, the renin-angiotensin system has been shown to play an important role in reproductive function, especially in follicular development. We found that administration of the ACE inhibitor affected the ovarian response in multiparous goats, with more visible follicles, and had no effect on oocyte quality or during embryonic development, thus being a possible alternative to improve goat reproductive response.

**Abstract:**

The aim of this work was to determine the effect of enalapril maleate administration, during oocyte recovery by serial laparoscopic ovum pick-up (LOPU), on the ovarian response and in vitro embryo production (IVP). Twenty cross-bred goats were allocated equally into two groups: Nulliparous and Multiparous. In each group, five animals were selected to receive daily doses of enalapril maleate during the hormonal protocol. Estrus was synchronized by a PGF2α analog, followed 48 h later by insertion of an intravaginal device with progesterone. Forty-eight hours after, a single dose of FSH/eCG was administered. The FSH/eCG doses were repeated three times, on every four day. Oocytes were recovered by LOPU 24 h after each FSH/eCG dose. Viable oocytes were matured in vitro, to be parthenogenetically activated and cultured for 72 h to the cleavage stage. The drug treatment increased the proportion of total follicles observed at LOPU (*p* < 0.01) in multiparous goats. In both parity groups, enalapril administration had no effect on the proportion or quality of oocytes recovered. Furthermore, the number of embryos cleaved was similar between the groups. Thus, enalapril maleate affected the ovarian response in multiparous animals only and had no effect on the oocyte quality or IVP.

## 1. Introduction

In recent years, goats have been used as animal models for developing modern reproductive biotechnologies, such as cloning and transgenesis [1]. However, regardless of recent advances, these biotechniques are still not very effective, with the main obstacle being the low quantity and quality of oocytes used as raw material for in vitro embryo production (IVP) [2].

Ovaries coming from slaughterhouses are the main source of oocytes for scientific research, providing a large quantity of egg cells at low costs. However, the quality of biological material from females with unknown genetic, health, and nutritional histories often limits its use and reduces the efficiency of reproductive biotechnologies. An alternative is to recover oocytes from live animals using laparoscopic ovum pick-up (LOPU), which provides high-quality oocytes [3] and allows repeatability in animals with breeding value. However, LOPU still shows variable efficiency depending on factors like the donor age and breed or aspects related to the technique used, such as the hormone stimulation protocol and the interval between collections (reviewed by [4]).

Many hormone stimulation protocols have been proposed with the objective of increasing the quantity and quality of oocytes recovered by LOPU section [5,6]. For many years, the decrease in ovarian response due to repeated hormonal treatments with short-term follicle-stimulating hormone (FSH) or equine chorionic gonadotropin (eCG) in goats has already been discussed as being a result of antibody formation against these hormones [7,8] as well as the duration of the intervals between LOPU procedures, by forming possible ovarian adhesions [9]. However, some recent studies have shown contrary results, showing no negative effect of repeated ovarian stimulation followed by LOPU on the quantity and quality of oocytes recovered with porcine FSH (pFSH) [4]. Sanchez et al. [10] reported that goats raised in the tropics, hormonally stimulated with pFSH, and then subjected to LOPU, using 7 repetitions every 2–3 weeks, maintained good ovarian response and embryo production. Additionally, Gibbons et al. [5], aiming to maximize IVP in goats and sheep, observed that stimulation using the one-shot regimen (FSH plus eCG) on every four day followed by LOPU, totaling three sessions, maintained the same pattern of ovarian response and embryo production.

Many intrinsic strategies have been adopted to improve the ovarian response in IVP programs. The modulation of the renin-angiotensin system (RAS) has been the objective of several researchers to improve reproductive biotechnologies, such as the response to estrus synchronization and superovulation and fixed-time artificial insemination (FTAI) [11,12]. In addition to their systemic effects on the regulation of blood pressure and body fluid homeostasis, RAS components were identified in the reproductive tract a long time ago [13,14] and there is evidence of their involvement in important reproductive processes, such as steroidogenesis, folliculogenesis, ovulation, and corpus luteum formation [15].

Costa et al. [16] reported higher levels of angiotensin (1–7) (Ang-(1–7), a RAS peptide) in the ovaries of rats during proestrus and estrus, suggesting that this peptide is involved in ovulation events. In addition, Viana et al. [17] described the in vitro perfusion of Ang-(1–7) in the ovaries of rats, which increased the concentrations of estradiol and progesterone and, consequently, the ovulation rate. According to Brosnihan et al. [18], an angiotensin-converting enzyme (ACE) inhibitor would prevent the conversion of angiotensin-1 into antiotensin-2, thereby increasing Ang-(1–7) production. In line with this, Costa et al. [11] reported higher concentrations of estradiol in sheep treated subcutaneously with an ACE inhibitor (enalapril maleate). Fernandes-Neto et al. [12] increased the pregnancy and twin birth rates in goats treated with enalapril maleate. In this context, an ACE inhibitor could be used to increase the efficiency of the ovarian response in IVP programs.

It is known that the expression and functionality of the systemic and ovarian RAS are species-specific and may vary according to age [19], and that data related to goats are still scarce. In addition, specific strategies are necessary to maximize IVP in this species, such as those for the collection of oocytes, collection interval, hormone protocol, and animal class. Therefore, the objective of this study was to evaluate the effect of enalapril maleate administration during a serial protocol of hormone stimulation and LOPU on the ovarian response, oocyte quality, and IVP in nulliparous and multiparous goats.

## 2. Materials and Methods

### 2.1. Location, Conditions, and Facilities

This experiment was conducted at the Experimental Farm of School Veterinary Medicine, Ceará State University (UECE), located in Guaiuba, CE. The equatorial region (4°2′23″ S and 38°38′14″ W) is characterized by a continuous photoperiod regimen, and has a warm, tropical, sub-humid climate with a mean annual rainfall and temperature of 904.5 mm and 27 ± 2 °C. The second part of the experiment, involving oocyte maturation and in vitro embryo production, was carried out in the Laboratory of Ruminants Nutrition at School Veterinary Medicine - UECE, Fortaleza, Ceará, Brazil. All procedures in this study were approved by the Ethics Committee in Animal Experimentation of the UECE (No. 6497021/2017, Ethics Committee for the Use of Animals, CEUA-UECE).

### 2.2. Animal Management

A total of 20 Anglo-Nubian cross-bred goats, were allocated to two groups, Nulliparous (n = 10) and Multiparous (n = 10) according to the order of parity. Each group was homogenous in body weight (BW) (26.5 ± 2.0 kg Nulliparous and 42.8 ± 2.1 kg Multiparous) and body condition scores (2.8 ± 0.1 overall mean) were determined, as proposed by Morand-Fehr [20], using a range from 1 (very thin) to 5 (obese), with progressive steps of 0.25. All animals received mineral salt and water ad libitum and were kept in collective stalls and fed with elephant grass, chopped and concentrated, to satisfy the nutrient requirements of breeding for non-dairy goats [21]. Prior to the experiment, females were selected from a farm herd and then submitted to a 45-day housing adaptation after receiving endo- and ectoparasite treatments.

### 2.3. Experimental Design

In each parity group (Nulliparous or Multiparous), five animals randomly chosen were segregated in two treatments (n = 10/treatment): control and enalapril maleate. The animals selected to enalapril maleate treatment had a stage of adaptation to the continuous administration of enalapril maleate, as described by Fernandes Neto et al. [12], with minor modifications (Figure 1). Briefly, the animals received crescent doses for three days (0.2 mg.kg^−1^ BW and 0.3 mg.kg^−1^ BW) of enalapril maleate for a total period of 6 days until reaching the dose stage at the beginning of estrus synchronization treatment of 0.4 mg.kg^−1^ BW, which was maintained throughout the experiment (11 days). The injectable solution of enalapril maleate was obtained from tablets (Enalapril, Teuto, Anápolis, Brazil) that were processed in a crucible and, after maceration, diluted and homogenized in 3 mL of 9% saline solution in a vortex for 2 min. Control treatment animals received the same volume of saline solution at 9% as placebo. Enalapril maleate and placebo were administered subcutaneously daily at 7 am.

### 2.4. Hormonal Treatment

Estrus synchronization and ovarian stimulation were performed according to Gibbons et al. [9]. Briefly, the ovarian status was synchronized in all goats by the intramuscular (i.m.) administration of 1 mL (0.075 mg) of a PGF2α analogue (Prolise; ARSA S.R.L., Buenos Aires, Argentina) and the insertion of an intravaginal progesterone insert (CIDR, InterAg, Hamilton, New Zealand) 48 h later. The intravaginal insert were removed after the last oocyte recovery. The follicular development was stimulated by the i.m. administration, at the same time, of a single dose of FSH (60 mg Folltropin; Vetrepharm, London, ON, Canada) plus a single dose of eCG (300 UI, Novormon 5000; Syntex, Argentina). The first FSH/eCG doses were administered 48 h after the dispositive insertion, and repeated every 4 days to complete 3 treatments (Figure 1).

### 2.5. Laparoscopic Ovum Pick-Up, Follicle Selection, and Oocyte Evaluation

Laparoscopic ovum pick-ups were performed 24 h after each FSH/eCG treatment and fasting for 36 h. At LOPU, the animals were anesthetized by proceeding under general anesthesia (xylazine i.m.: 0.2 mg/kg of Anasedan 2%, Ceva, São Paulo, Brazil; and ketamine iv: 2 mg/kg of Dopalen 10%, Ceva, São Paulo, Brazil) and local anesthesia applied in the surgical field (Lydocaine: Lidovet, BRAVET, Rio de Janeiro, Brazil). The anesthetized animals were immobilized in a cradle in the dorsal position for laparoscopic surgical procedures. Thereafter, an endoscope (5 mm 0°, Stryker, San Jose, USA) was inserted into the abdominal cavity, the ovaries were manipulated and fixed by the use of an atraumatic clamp (5 mm, EXATECH, Porto Alegre, Brazil), and follicular fluid aspiration were performed by a follicular aspiration needle for small ruminants (20G, WTA, São Paulo, Brazil) connected to a Falcon tube under controlled vacuum (35 mm Hg). The follicles present in both ovaries of each female were categorized according to their diameter as small (2 mm), medium (2–3 mm), or large follicles (>4 mm) and, subsequently, aspirated. The follicular fluid of each animal was aspirated separately into collection tubes containing Dulbecco’s phosphate buffered saline, D-PBS (Nutricell, Campinas, SP, Brazil), at 38 °C, supplemented with 5% fetal calf serum, FCS (Sigma Chemical Co., St Louis, MO, USA), antibiotics (100 IU/mL penicillin plus 0.1 mg/mL and streptomycin (Sigma Chemical Co., St Louis, MO, USA), and 0.05 mg/mL heparin (Liquemine, Campinas-SP, Brazil). The follicular fluid sediment was used to recover the cumulus oocyte complexes (COCs) that were classified according to Baldassarre et al. [22] with minor modifications. Briefly, assessment of the quality of COCs was based on visual criteria with the use of a stereomicroscope (SMZ-645; Nikon, Tokyo, Japan) according to three different classes: GI (with multilayered compact cumulus and evenly homogeneous oocyte cytoplasm); GII (with 2–3 layers of cumulus cells and evenly homogeneous oocyte cytoplasm); GIII (with 1 layers of cells of the cumulus, partially or totally denuded with homogeneous oocyte cytoplasm); or degenerated (denuded with heterogeneous oocyte cytoplasm).

### 2.6. In Vitro Maturation (IVM), Parthenogenetic Activation and In Vitro Culture (IVC)

All COCs, segregated as viable by grade (G-I to G-III) according to morphological aspects and experimental group, were in vitro-matured at 38.5 °C and 5% CO2 in air, in Petri dishes (Corning, USA) under mineral oil containing drops with 100 µL maturation medium, which consisted of TCM 199 medium supplemented with 0.022 µg/mL sodium pyruvate, 10,000 IU penicillin, 10,000 µg/mL streptomycin sulfate, 10% FCS, 10 ng/mL EGF, 5 µg/mL FSHp (Folltropin; Bioniche, Belleville, Ontario, Canada), 10 µg/mL LH (Lutropin; Bioniche), 1 µg/mL 17β-estradiol, and 100 µM cysteamine. After 24 h in vitro-matured oocytes were parthenogenetically activated by exposure to 5 mM ionomycin for 5 min, followed by incubation in 2 mM 6-dimethylaminopurine in G1 (Vitrolife) medium for 4 h. After activation, presumed zygotes were cultured in medium G1 in incubator at 38.5 °C and 5% CO_2_ in a humidified atmosphere for 72 h. Subsequently, embryos were incubated in 100 µL droplets of TCM199–HEPES with 10 M Hoechst 33342 stain (Sigma, Deisenhofen, Germany) at 38.5 °C for 30 min and individually examined under a fluorescence microscope (Nikon, Eclipse 80i, Tokyo, Japan) to visualize DNA of all cells to determine cleavage rates.

### 2.7. Statistical Analysis

Data were subjected to analysis of variance (ANOVA) using the general linear models (GLM) procedures (Statistica v. 13.4.0.14, TIBCO Software, Inc., Palo Alto, CA, USA). The model included the main factors: treatment (Control, Enalapril-maleate), group of parity (Nulliparous and Multiparous), number of serial LOPU (1, 2, and 3), and interaction treatment vs. group of parity, and treatment vs. number of serial LOPU. Pairwise comparisons were performed by the Student’s t-test or Newman–Keuls test. Data were log-transformed (logx10).

## 3. Results

The main results are shown in Table 1 and Table 2. No significant effects (*p* > 0.05) of enalapril maleate treatment on the two parity groups were observed for all parameters analyzed. The use of repeated oocyte retrieval by LOPU significantly affected (*p* < 0.01) the proportion of small follicles (Table 1) and degenerate oocytes (Table 2). There was an increase (*p* < 0.01) in the number of follicles (Figure 2) between the first and subsequent LOPUs, and a decrease (*p* = 0.03) in the number of degenerate oocytes in the last recovery procedure compared with that in the first one (Figure 3).

In relation to the number of large follicles and total follicles (Table 1), there was a significant interaction (*p* < 0.01) between the enalapril maleate treatment and the parity group. The results of this interaction for the total follicles are represented in Figure 4. Multiparous animals treated with enalapril maleate had a higher number (*p* < 0.01) of follicles than that in the untreated control animals and the treated nulliparous animals (*p* = 0.02).

## 4. Discussion

The presence of the RAS has been reported in different reproductive structures of rats, such as granulosa cells, corpus luteum, and oocytes [23], and in bovine follicular fluid [24]. In the RAS complex, higher concentrations of the Ang-(1–7) peptide were reported in rats during the follicular phase [16], suggesting the participation of this molecule in important processes, such as folliculogenesis, steroidogenesis, corpus luteum formation, and oocyte maturation.

RAS modulation through ACE inhibition is justified by the involvement of the Ang-(1–7) peptide in the secretion of hormones, such as estradiol and progesterone, and in the ovulation rate. For this reason, it has been used in goats to maximize the efficiency of reproductive biotechniques in the field, such as estrus synchronization [11], superovulation treatments [25], and FTAI [19].

The results confirmed the efficiency of enalapril maleate as an ACE inhibitor, considering the significant stimulation of the ovarian response in the pluriparous animals. Recent studies have reported similar activity in different species and using different ACE inhibitor drugs. Viana et al. [17], Costa et al. [11], and Fernandes-Neto et al. [12] reported increased ovarian activity from using enalapril maleate in rats, sheep, and goats through the increase of parameters, such as the ovulation rate, prolificacy, or estradiol levels. Nielsen et al. [26] reported increased estradiol concentration in cows treated with captopril, another molecule with ACE inhibition action.

Although the presence of several RAS components in the reproductive tract has been confirmed [16,27], little is known about variations in this system due to reproductive history or age. Many studies have reported the effect of reproductive age on the ovarian response to superovulation in ruminants, with most of them comparing prepubertal and adult animals [28,29]. However, it is known that young or prepubertal animals are responsive to exogenous gonadotrophin stimulation ([28,30], but at a lower intensity than that in adult and multiparous animals, and mainly in regard to ovulation rate [30,31] and embryo production [32], which may justify the low response to enalapril maleate found in the nulliparous animals in the present study.

The use of the serial superovulation/LOPU protocol was effective in terms of the ovarian response and oocyte quality, as also reported by Gibbons et al. [9] for the same hormone protocol and interval between aspirations. According to Baldassare et al. [33], repeated oocyte recoveries in a short interval can maximize IVP, being more efficient than traditional superovulation followed by embryo transfer. In addition, our study showed an increased follicular response between successive collection procedures. The maintenance of the hormone treatment response between sessions corroborates the current concept of anti-eCG antibody production, a mechanism that usually results in a decreased ovarian response in the donors [34]. According to Gibbons et al. [9], a possible explanation may be the short interval between stimulations and the induced synchrony between the hormone protocol and follicular status of the follicular wave.

Many authors have reported an association between LOPU repetition and increased variability in the individual response of donors [9,35]. On the other hand, Sanchez et al. [10] did not find any decrease in the oocyte recovery rate in repeatedly stimulated goats with collection at every 2 weeks. In our experiment, five animals showed omental and ovarian adhesions in the third LOPU session. However, the formation of adhesions in short-interval LOPU programs is a possible phenomenon that should be prevented during the process of fixing and washing the ovaries with heparin after aspiration.

With regard to the IVP, the efficiency obtained (65.6% mean cleavage rate) corroborated that of specific literature (reviewed by [2]), which reported cleavage rates of between 60% and 70% in goats. The lack of any evident ACE inhibitor action on the initial embryonic development, as shown in our results, raises some questions related to the real role of the RAS in IVP. Contradictory evidence is present in the literature with regard to the effects of the age or parity of oocyte donors on the oocyte quality or IVP in ruminants (cattle: [36]; goats: [37]; and sheep: [38]). However, we know that old animals produce lower quality oocytes for in vitro development, mainly due to the lower number of cumulus cell layers [39].

## 5. Conclusions

In summary, enalapril maleate administration during a serial hormonal protocol in stimulated goats efficiently increased the ovarian response of the animals with an extensive reproductive history and the quality of oocytes recovered by laparoscopy. However, despite the significant increase of follicles recorded in the multiparous animals, the application of the drug did not result in positive effects on the oocyte retrieval procedure in terms of the quantity or quality of the egg cells and the proportion of embryos produced in vitro. This absence of efficient synergy between the LOPU procedure and the administered enalapril maleate suggests that this drug is ineffective in goats, at least under the experimental conditions proposed in the present investigation.

## Figures and Tables

**Figure 1 animals-09-00868-f001:**
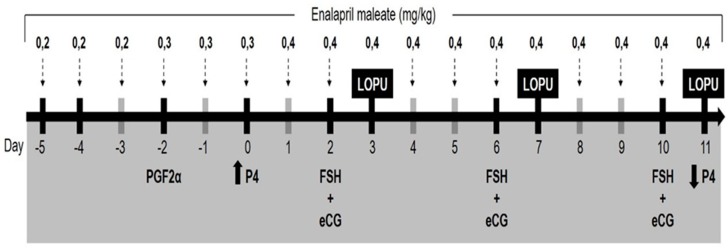
Timeline of enalapril maleate administration, estrus synchronization, and ovarian stimulation in goats oocyte donors by serial laparoscopic ovum pick-up (LOPU) (protocols adapted from Gibbons et al. [9].

**Figure 2 animals-09-00868-f002:**
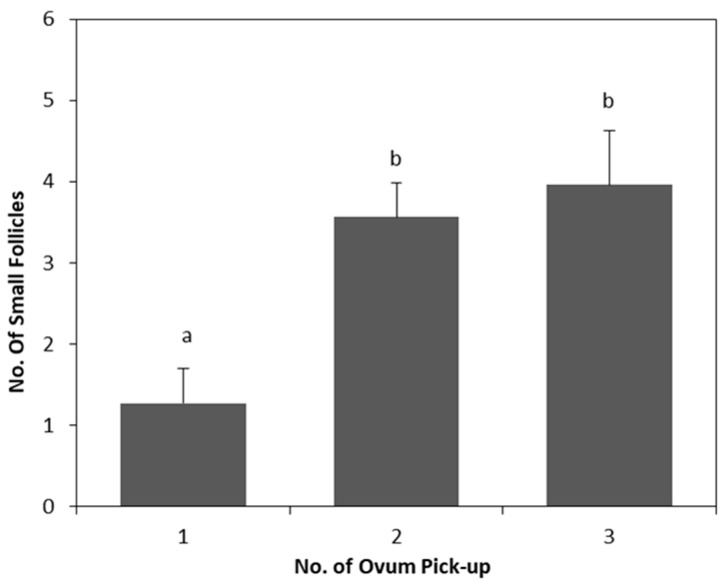
Effect of serial LOPU on number of small follicles from hormonal stimulated goats. a,b: bars with different letters between No. of LOPU differ (*p* < 0.05). Values are expressed as mean ± standard error of the mean.

**Figure 3 animals-09-00868-f003:**
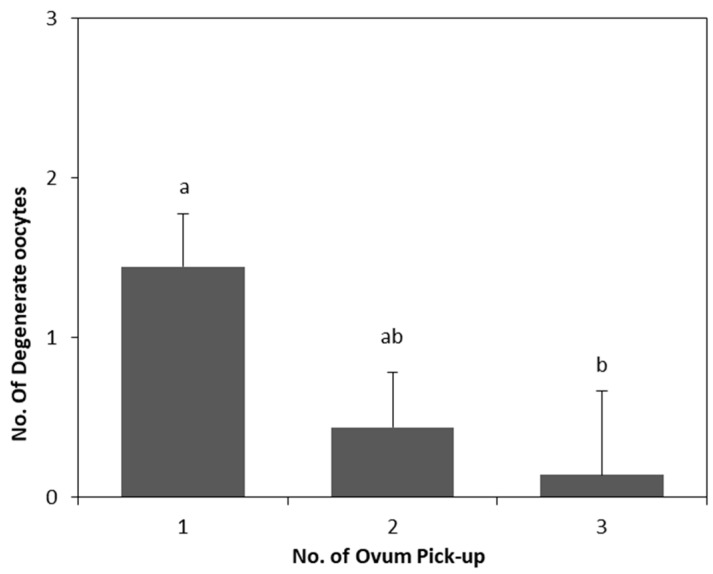
Effect of serial LOPU on number of degenerate oocytes recovered from hormonal stimulated goats. a,b: bars with different letters between No. of LOPU differ (*p* < 0.05). Values are expressed as mean ± standard error of the mean.

**Figure 4 animals-09-00868-f004:**
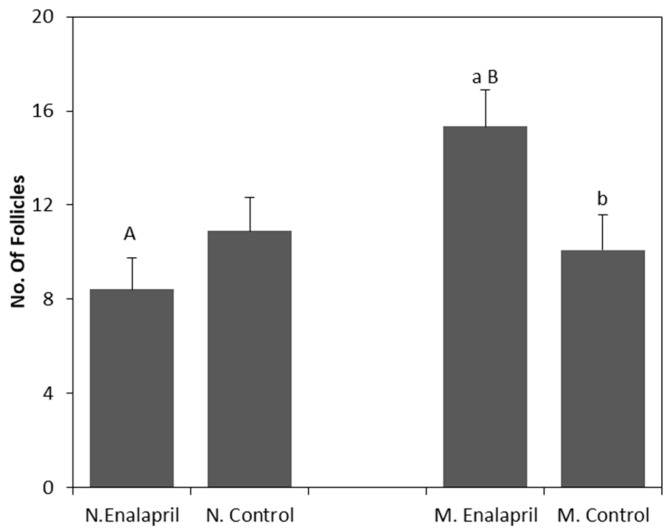
Effect of the interaction Treatment vs. Group of parity on number of total follicles from Nulliparous and Multiparous hormonal stimulated goats treated with enalapril maleate. a,b: bars with different letters between Treatment differ (*p* < 0.05). A,B: bars with different letters between Group of parity differ (*p* < 0.05). Values are expressed as mean ± standard error of the mean.

**Table 1 animals-09-00868-t001:** Ovarian response from Nulliparous and Multiparous hormonal stimulated goats treated with enalapril maleate. Values are expressed as mean ± standard error of the mean.

Attributes	Treatment			*p*-Value				
	Control	Enalapril Maleate	Mean	Treatment	Group	No. LOPU	T vs. G	T vs. L
No. of goats exposed	10	10						
*Ovarian response*								
Small Follicles, n	3 ± 0.4	3 ± 0.4	3 ± 0.3	0.55	0.09	0.01	0.21	0.85
Medium Follicles, n	4 ± 0.9	5 ± 0.9	5 ± 0.6	0.46	0.16	0.06	0.10	0.50
Large Follicles, n	3 ± 0.7	4 ± 0.7	4 ± 0.3	0.31	0.86	0.73	0.01	0.47
Total Follicles, n	10 ± 1.3	12 ± 1.2	11 ± 0.8	0.22	0.11	0.63	0.01	0.33

ANOVA results for the effects of Treatment (T), Group of Parity (G), and Number of serial LOPU (L).

**Table 2 animals-09-00868-t002:** Oocyte quality and in vitro embryo yield from Nulliparous and Multiparous hormonal stimulated goats treated with enalapril maleate. Values are expressed as mean ± standard error of the mean.

Attributes	Treatment			*p*-Value				
	Control	Enalapril Maleate	Mean	Treatment	Group	No. LOPU	T vs. G	T vs. L
No. of goats exposed	10	10						
*Oocyte and embryo*								
Viable oocytes, n	7 ± 1.1	7 ± 1.0	7 ± 0.6	0.50	0.43	0.76	0.07	0.27
Degenerate oocyte, n	1 ± 0.4	1 ± 0.4	1 ± 0.2	0.35	0.25	0.04	0.87	0.53
Recovery rate, % (n/n)	75.6 (167/221)	62.2 (153/246)	66.4 (320/467)	0.54	0.71	0.73	0.18	0.20
Embryo, n	5 ± 0.9	5 ± 0.9	5 ± 0.9	0.15	0.11	0.52	0.53	0.27
Cleavage rate, % (n/n)	59.3 (99/167)	71.9 (110/153)	65.6 (209/320)	0.05	0.30	0.72	0.77	0.21

ANOVA results for the effects of Treatment (T), Group of Parity (G), and Number of serial LOPU (L).

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
