# Peer review of "Effect of Continuous Administration of Enalapril Maleate on the Oocyte Quality and In Vitro Production of Parthenote Embryos in Nulliparous and Multiparous Goats Undergoing Serial Laparoscopic Ovum Pick-Up"

_animals, 2019, doi:10.3390/ani9110868_

Round 1
Reviewer 1 Report
This manuscript is well written and presents some interesting data demonstrating an improved protocol for the production of viable oocytes in goats a notoriously difficult species to work with. I have only a few minor comments.
Line 19-20 Is advance reproductive age the best term here given that it suggests they were old rather than multiparous. Also indicate what the effect was.
Line 55. Explain what you mean by "intervals between". I am assuming it means a lengthy interval between treatments.
Line 51-62. It is not clear why an antibody response should not occur in the newer studies vs the older studies? Some explanation is needed.
Author Response
Response to Reviewer 1 Comments
We appreciate the review of our manuscript. Modifications suggested by the reviewer have been incorporated into the revised version of the text, and the specific comments concerning reviewer’s queries are shown below.
Point 1: Line 19-20 Is advance reproductive age the best term here given that it suggests they were old rather than multiparous? Also indicate what the effect was.
Response 1: We appreciate this observation. Thus, we rewrote the sentence giving a better understanding of the information replacing the sentence by "multiparous" and adding the effect found (Line 21).
Point 2: Line 55. Explain what you mean by "intervals between". I am assuming it means a lengthy interval between treatments.
Response 2: To improve understanding of this sentence, we replace the confusing text by “as well as the duration of the intervals between LOPUs procedures, by forming possible ovarian adhesions” (Line 56)
Point 3: Line 51-62. It is not clear why an antibody response should not occur in the newer studies vs the older studies? Some explanation is needed.
Response 3: We understand the reviewer's concern, so we rewrote the paragraph giving a better understanding of what was cited (Line 52-65)
Reviewer 2 Report
I do not have any suggestions for this manuscript. I feel the experiment is justified, the experimental n is adequate, and the statistical methods are appropriate. The results are well presented and interpreted.
Author Response
Response to Reviewer 2 Comments
We appreciate the review of our manuscript. As there were no suggestions made by the reviewer, only few details were entered into the manuscript according to the suggestions of the reviewer 1.